# A Review of Biomarkers of Cardiac Allograft Rejection: Toward an Integrated Diagnosis of Rejection

**DOI:** 10.3390/biom12081135

**Published:** 2022-08-18

**Authors:** Guillaume Coutance, Eva Desiré, Jean-Paul Duong Van Huyen

**Affiliations:** 1Department of Cardiac and Thoracic Surgery, Cardiology Institute, Pitié-Salpêtrière, Assistance Publique-Hôpitaux de Paris (AP-HP), Sorbonne University Medical School, 75013 Paris, France; 2INSERM UMR 970, Paris Translational Research Centre for Organ Transplantation, University of Paris, 75013 Paris, France; 3Pathology Department, Hôpital Necker, Assistance Publique-Hôpitaux de Paris (AP-HP), Université de Paris, 149 rue de Sèvres, CEDEX 15, 75743 Paris, France

**Keywords:** heart transplantation, allograft rejection, biomarker, molecular biology

## Abstract

Despite major advances in immunosuppression, allograft rejection remains an important complication after heart transplantation, and it is associated with increased morbidity and mortality. The gold standard invasive strategy to monitor and diagnose cardiac allograft rejection, based on the pathologic evaluation of endomyocardial biopsies, suffers from many limitations including the low prevalence of rejection, sample bias, high inter-observer variability, and international working formulations based on arbitrary cut-offs that simplify the landscape of rejection. The development of innovative diagnostic and prognostic strategies—integrating conventional histology, molecular profiling of allograft biopsy, and the discovery of new tissue or circulating biomarkers—is one of the major challenges of translational medicine in solid organ transplantation, and particularly in heart transplantation. Major advances in the field of biomarkers of rejection have paved the way for a paradigm shift in the monitoring and diagnosis of cardiac allograft rejection. We review the recent developments in the field, including non-invasive biomarkers to minimize the number of protocol endomyocardial biopsies and tissue biomarkers as companion tools of pathology to refine the diagnosis of cardiac rejection. Finally, we discuss the potential role of these biomarkers to provide an integrated bio-histomolecular diagnosis of cardiac allograft rejection.

## 1. Introduction

Heart transplantation (HTx) remains the most valuable therapeutic option for patients with end-stage heart failure [1,2,3]. Despite major advances in immunosuppression, allograft rejection remains an important complication after heart transplantation, and it is associated with increased morbidity and mortality [4,5,6]. While the prevalence of acute cellular rejection (ACR) decreased over time from the initiation of the transplantation programs due to the improvements in immunosuppression [7,8], antibody-mediated rejection (AMR) is recognized as a major risk factor for all-cause mortality [9,10], cardiovascular mortality [11,12], and various types of allograft injury, including systolic dysfunction [5,13], restrictive physiology [14], and cardiac allograft vasculopathy [15,16,17]. The pathologic evaluation of myocardial tissue taken during an endomyocardial biopsy (EMB) remains the gold standard approach to monitor and diagnose cardiac allograft rejection. International guidelines recommend the performance of endomyocardial biopsies in two clinical scenarios: (i) For-cause EMB when there is clinical suspicion of allograft rejection (symptoms of heart failure, allograft dysfunction, arrythmias, etc.) and (ii) protocol EMB (i.e., in asymptomatic patients with normal cardiac explorations), at least during the high-risk first year post-transplantation period, to detect allograft rejection at a subclinical state since multiple reports have highlighted the severity of rejection-associated allograft dysfunction [5,13,18]. However, this invasive strategy suffers from important limitations. First, in the current era of potent immunosuppressive drugs, the prevalence of rejection based on protocol EMB has been reported to be very low [7,8]. Therefore, this invasive strategy is increasingly questioned due to inherent complications, costs, and impacts on patients’ quality of life [19]. EMB as a screening test does not fulfill all the principles of screening recommended by the World Health Organization. Alternative non-invasive strategies are urgently needed. Non-invasive circulating biomarkers are required to minimize the use of protocol EMB since no robust clinical prediction model can accurately discriminate between patients with and without rejection. Second, the pathological evaluation remains an imperfect gold standard due to sample bias, high inter-observer variability, and limited international pathology working formulations based on arbitrary cut-offs that oversimplify the complex landscape of cardiac allograft rejection [20,21,22]. New molecular biology developments based on myocardial tissue suggest that this approach may be a valuable tool by providing additional information on the involved pathways and the molecular activity.

Overall, there is a strong and growing literature suggesting a crucial role of invasive and non-invasive biomarkers to better monitor, detect, and refine the diagnosis of allograft rejection after heart transplantation. In this article, we review existing data concerning the use of (i) circulating non-invasive biomarkers of allograft rejection to reduce the number of protocol EMB and (ii) tissue biomarkers as a companion tool for the pathologist to better characterize allograft rejection.

## 2. Non-Invasive Biomarkers of Rejection to Minimize Routine Endomyocardial Biopsies

During the last two decades, important resources have been allocated to the search for an accurate non-invasive biomarker of allograft rejection. These biomarkers can be classified into two categories: Those reflecting allograft injury and those reflecting the inflammatory and allo-immune processes underlying allograft rejection. Due to the potential consequences of missing and not treating acute cardiac allograft rejection, these biomarkers are required to be highly sensitive for rejection even at the cost of low specificity. Despite dozens of promising studies and candidate biomarkers, only two have been approved by the Food and Drug Administration (FDA) and applied in routine clinical practice, mostly in North America (AlloMap and donor-derived cell-free DNA [dd-cfDNA]) [23]. The overall literature on non-invasive biomarkers is often subject to bias, particularly selection bias (sick versus well, severe historic rejection compared with pristine biopsies) and limited challenge bias (no or limited multivariable analysis included other validated risk markers/factors of rejection, e.g., donor-specific antibodies [DSA]) [24,25]. The results of numerous retrospective case-control studies have never been confirmed in large prospective studies of unselected patients, probably reflecting the over-selection of the cases (sickest among the sickest) and controls (uncomplicated long-term survivors). We propose the steps described in Table 1 to develop a new biomarker before a clinical application in the field of cardiac rejection. Most biomarkers tested have ended early in their development phase and are far from clinical use. Rather than listing all candidate biomarkers, we focus our review on two biomarkers of allograft injury (troponin and dd-cfDNA) and two circulating biomarkers reflecting the inflammatory and allo-immune processes underlying allograft rejection (AlloMap and microRNA [miRNA]).

### 2.1. Non-Invasive Biomarkers of Allograft Injury

#### 2.1.1. Troponins

Troponins are structural proteins of the myocardium, expressed almost exclusively in the heart. They consist of three subunits: Troponin T, C, and I. Troponin T comprises two tissue subunits, one of which is myocardium-specific, cTnT. cTnT is a biomarker mainly used in the diagnosis of acute coronary syndromes and myocarditis. However, its elevation in the blood does not reflect the underlying mechanism, but only the existence of myocardial necrosis. Therefore, the use of this circulating marker of myocardial injury, which is a rapid, non-invasive, and routinely performed assay, has been evaluated as a potential component of the strategy for the non-invasive monitoring of cardiac allograft rejection. Multiple studies, mostly retrospective single-center studies, have reported conflicting results concerning the association between troponins and rejection [26,27]. They have been summarized in a recent systematic review [28]. Authors found wide variation in diagnostic performance, with cTn assays demonstrated sensitivity between 8 and 100% and specificity between 13 and 88% for detection of ACR. The positive predictive value (PPV) was low but the negative predictive value was high (79–100%). High sensitivity cTn assays had greater sensitivity and negative predictive value than conventional cTn assays for detection of ACR. However, the analysis and the comparison between studies is challenging due to the significant heterogeneity concerning their methodology and by their cross-sectional design.

In a longitudinal study including unselected HTx recipients and applying a rigorous statistical approach, the temporal evolution of Troponin T did not predict the occurrence of ACR both in the early and late course of the first year after HTx [27].

Additionally, cTn assay should probably not be used as an isolated non-invasive biomarker of rejection since (i) only ACR ≥ 2R has been evaluated thus neglecting AMR, (ii) only severe rejections with significant allograft injury are expected to increase troponin plasma level, and (iii) no threshold has been defined (importance of time post-transplant and baseline troponin level for each patient).

#### 2.1.2. Donor-Derived Cell-Free DNA

Healthy individuals have a small amount of cfDNA corresponding due to physiological cell death. The majority of this cfDNA is released from hematopoietic cells, with <1% release from the heart. The cfDNA is increasingly used in the medical field in oncology and pre-natal diagnosis.

The dd-cfDNA, detected in the blood of transplant recipients, has been proposed as a non-invasive marker of graft injury [29,30]. Early dd-cfDNA studies had been based on the hypothesis that acute rejection causes cell death in the allograft, which leads to increased levels of dd-cfDNA in the recipient’s bloodstream. Major technological advances in the field of DNA sequencing have allowed widespread use of these techniques. Even without sequencing a donor’s DNA, it is possible to accurately differentiate the origin of two human DNA sources through shotgun sequencing of hundreds of selected single nucleotide polymorphisms that may vary between the donor and the recipient. Differently from many biomarkers in the field, there is strong and growing literature supporting the use of dd-cfDNA as a reliable marker of cardiac allograft injury (Table 1). Not only is there a clear biological plausibility, but also the test is highly reproducible with known kinetics after transplant and demonstrated stability over time in healthy subjects. The dd-cfDNA has been associated with allograft rejection in case-control studies [31] as well as multicenter cross-sectional studies including unselected patients [32]. A key National Institutes of Health (NIH)-funded large multicenter study with a longitudinal follow-up of patients has recently been published by Hannah Valantine and her team. They found a strong correlation between dd-cfDNA and rejection (particularly AMR), but also demonstrated a rise in dd-cfDNA before the occurrence of biopsy-proven rejection and a drop following the treatment of rejection, findings that suggest the potential clinical utility of this biomarker. The diagnosis accuracy was high, with an area under the curve above 0.85 [33]. The 2022 updated guidelines from the International Society for Heart and Lung Transplantation (ISHLT) support the use of dd-cfDNA as a valid non-invasive biomarker to rule out rejection. As a result, some centers, mostly from North America, have adopted dd-cfDNA for rejection surveillance and to reduce the number of EMB during the period of 3 months to 1 year post-transplantation [34]. However, the clinical application of dd-cfDNA is still limited by costs and availability of platforms, particularly outside the US [32,33].

The future developments of dd-cfDNA in heart transplantation include: (i) An ongoing multicenter randomized clinical trial (DETECT trial, NCT05081739, non-inferiority trial comparing the use of dd-cfDNA to the current standard of care based on protocol EMB, primary endpoint: Treated rejection with graft dysfunction, treated rejection without graft dysfunction, graft dysfunction, redo heart transplantation, and death), and (ii) in-depth analysis of dd-cfDNA length and the absolute number of dd-cfDNA in opposition to the relative proportion currently considered.

### 2.2. Non-Invasive Biomarkers Reflecting the Inflammatory and Allo-Immune Processes Underlying Allograft Rejection

#### 2.2.1. AlloMap

Allomap is a non-invasive test based on gene-expression profiling (GEP) of peripheral blood mononuclear cells, that has been developed for the diagnosis of ACR in HTx. Transcripts of interest reflect diverse immunoregulatory pathways from a variety of immune and non-immune cells. After broad microarray analyses and among 252 pre-selected candidate genes, 11 were finally significantly associated with ACR (PDCD1 [T lymphocyte activation], SEMA7A [T lymphocyte], ARHU [unknown], MARCH8 [hematopoiesis], WDR40 [hematopoiesis], ITGAM-IL1R2-FLT3 [steroid responsive], G6B-PF4 [platelet], and ITGA4 [T lymphocyte migration]). A score ranging from 0 to 40 was developed and validated. In two validation cohorts, GEP appeared to detect accurately ACR ≥ 2R on the concomitant EMB. Patients > 1 year post-transplant with scores below 30 were unlikely to have grade ≥ 2R rejection (negative predictive value = 99.6%) [35]. In the IMAGE study, 602 recipients ≥ 6-month post-transplant were randomized between the GEP and the standard protocol based on protocol EMB to monitor rejection. The protocol was amended during the study period to change the threshold for a mandatory EMB from 30 to 34. The GEP strategy (i) appeared to be non-inferior for the primary combined endpoint including first occurrence of rejection with hemodynamic compromise, graft dysfunction due to other causes, death, or retransplantation and (ii) was associated with a lower number of EMB performed [36]. Another trial that randomized patients at 2-month post-transplant found similar results (a lower score threshold [score ≥ 30] was applied between 2- to 6-month post-transplant) [37]. The European observational study CARGO-II evaluated the performance of GEP for patients with a recent HTx. For ≥2–6 and ≥6 months post-HTx, GEP score performance (AUC-ROC = 0.70 and 0.69) was similar to the CARGO study results [38].

This test received approval from the FDA in 2008 and the CE marking in 2011. International guidelines support the use of GEP as a non-invasive biomarker to monitor rejection. However, the use of this screening test as an alternative to the EMB in asymptomatic patients suffers from several limitations: (i) AlloMap comprises a limited number of genes expressed by a limited number of cells compared with various pathways and cell types implicated in the pathophysiology of rejection, (ii) GEP has only been evaluated to rule out severe ACR, excluding mild episodes of ACR and AMR, (iii) its high negative predictive value to rule out ACR must be interpreted in the context of a low and declining incidence of significant ACR in HTx, (iv) GEP is not specific of rejection and its positive predictive value is low, (v) the gene signature found in the peripheral blood has never been validated in allograft biopsies, (vi) its use remains limited in Europe due to costs, limited availability, and the absence of a specific economic evaluation.

#### 2.2.2. Circulating Micro-RNA

Small and non-coding RNAs called miRNAs have been shown to be involved in gene expression regulation. Although miRNAs are known to be involved in many biological processes, such as development, cell proliferation, differentiation, apoptosis, and oncogenesis, increasing evidence suggests that they may play a critical role in the regulation of immune cell development and in the modulation of innate and adaptive immune responses. Several studies have reported a potential causative role of miRNAs in the pathophysiology of cardiac allograft rejection and distinct miRNA profiles in EMB from patients with or without rejection [39,40,41,42]. These studies have been summarized in a comprehensive review [43]. Consequently, there has been increased interest in miRNAs in the field of non-invasive biomarkers of organ allograft rejection.

In a multicenter retrospective case-control study, among 17 pre-selected candidate miRNAs, four (miR-10a, miR-31, miR-92a, and miR-155) showed differential tissue and serum expression between rejection and normal heart allografts. There were strong correlations between tissue and serological expression of these four miRNAs. Assessment of these miRNAs in patient sera permitted very high accuracy discrimination between patients with and without allograft rejection. Since then, several observational studies, mostly case-control and single-center studies, have identified various miRNAs as non-invasive biomarkers of rejection, mostly ACR: miR-144-3p [44], miR-181a-5p [45], miR-142-3p and miR-101-3p [46], and miR-29c-3p and miR-486-5p [47]. Despite the broad analysis of unselected miRNA in most of the recent studies, the heterogeneity in the type of miRNA identified across studies is striking and suggests a high degree of variability in the results. To date, there have been no prospective studies with a longitudinal follow-up of unselected patients. The results of a large NIH study and a French prospective study are awaited (NCT02672683).

Several studies have reported on the potential interest of miRNA profiling of myocardial tissue. In a study combining human myocardial tissue (case-control) and a murine model of ACR, and performing an unsupervised analysis of miRNA (miRNA expression profiling on the nCounter^®^ platform (NanoString Technologies, Seattle, WA, USA), miRNA profiling revealed that human and murine ACR share significant dysregulation of immune genes. Inflammatory miR-155 was the most differentially expressed between ACR and controls in both human and animals. Importantly, the authors of this study demonstrated that absence or pharmacological inhibition of miR-155 attenuated ACR, demonstrating the causal involvement and therapeutic potential of miRNAs.

Although these studies have provided interesting and promising results, they still cannot support the use of miRNAs in the clinical field. Large multicenter prospective studies with a longitudinal follow-up of unselected patients are still lacking. These studies are mandatory to minimize bias related to biomarker studies and to confirm the potential interest of this biomarker.

Recent research suggests that circulating extracellular vesicles may be a valuable non-invasive biomarker of rejection. They are raising considerable interest as they are easily detectable in blood and contain a specific set of nucleic acids, proteins, and lipids reflecting pathophysiologic conditions. Two recent retrospective case-control studies reported promising results that should be further evaluated and validated in unselected longitudinal cohorts [48,49].

### 2.3. Individual Risk Stratification of Allograft Rejection

An alternative data-driven approach is to develop predictive models aimed at stratifying the individual risk of rejection. Several models have been developed but are not routinely applied in clinical practice [50,51]. In addition to their poor statistical performance, a major limitation has been the development of patient-based models rather than biopsy-based models. In the latter, the pre-test probability of rejection can change over time for any given patient according to fluctuation in risk factors. Improving the stratification of the risk of AMR might be an important step toward individualized monitoring of rejection.

Based on large, international and deeply phenotyped cohorts, we recently identified five independent predictive variables associated with biopsy-proven AMR during the first year post-transplant and built a clinical risk prediction model for AMR. It showed excellent discrimination in the US test set and in an independent external validation cohort. Simulation analyses suggested that individualizing the EMB protocol according to the predicted probability of AMR may safely reduce the number of EMB performed. A user-friendly web-based interface allowing an open access evaluation of the pre-test probability of AMR on any EMB performed during the first year post-transplant was built (https://transplant-prediction-system.shinyapps.io/antibody_mediated_rejection_risk_model/, accessed on 17 August 2022).

## 3. Molecular Biology as a Companion Tool to Refine the Diagnosis of Rejection

Non-invasive biomarkers of rejection should mostly be seen as a first step to rule out rejection in asymptomatic patients, with the goal of minimizing the use of routine EMB in low-risk situations. However, due to the complex pathophysiology of allograft rejection and the shared pathways and involved cells between ACR and AMR, circulating non-invasive biomarkers are unlikely to provide accurate and valuable information concerning the type of ongoing rejection. On the other hand, molecular biology analysis of the myocardial sample may be an interesting companion tool for the pathologist to refine the diagnosis of rejection and to analyze the intensity of rejection-associated molecular activity.

The accuracy of pathology to assess accurately the status of the cardiac allograft is limited by sampling errors and reproducibility issues among pathologists [20]. These issues are mainly related to an arbitrary diagnostic scheme and evaluation of rejection based on a semi-quantitative scale, which oversimplifies a complex phenotype [21,22]. The histological gold standard has limitations particularly for the recognition of complex and mixed rejections, with direct consequences on the choice of treatments and the management of patients. This need for diagnostic improvement has been highlighted by government agencies, transplantation societies, and international transplantation consortiums.

In kidney transplantation, molecular profiling strategies using pangenomic approaches to assess graft biopsies have made it possible to define signatures of pathogenic transcripts relevant for the diagnosis of allograft rejection. These molecular signatures have shown an association with rejection grades, organ function, and long-term graft outcome, and they are now included in the algorithm to define rejection. In heart transplantation, preliminary studies suggest that cardiac rejection is a more complex and heterogeneous disease in terms of its molecular representation than defined by histology [52], while helping to clarify the importance of certain lesions in the disease activity [52,53]. In this section, we review the evidence supporting the interest of various molecular biology approaches to refine the diagnosis of cardiac allograft rejection.

### 3.1. Pangenomic Approaches

Microarray technology (Affymetrix^®^, Thermo Fisher Scientific, Waltham, MA, USA) has been used to define the molecular pathways involved in cardiac allograft rejection [52,54,55]. These were mainly case-control studies, comparing between the transcriptome of EMB with ACR or AMR and EMB without rejection, followed by canonical pathway analysis of the most differentially expressed transcripts between the different histological diagnoses. As an example, EMB with AMR showed high expression of interferon-gamma-related transcripts, endothelial transcripts induced by microcirculation activation and damage, and monocyte/macrophage and natural killer (NK) cell transcripts [55]. However, it should be noted that the cases included in these studies were selected based on histological diagnosis, and thus on established tissue damage. Therefore, these analyses are mainly focused on the effector mechanisms of rejection, possibly after multiple inflammatory amplification loops. In the context of the recent description of new mechanisms of allograft rejection [56,57], it would be particularly interesting to perform a longitudinal molecular follow-up to redefine molecular signatures both before the establishment of histological lesions and after the treatment of rejection. To date, no longitudinal transcriptomic studies have been performed in heart transplantation. Therefore, information regarding histomolecular correlations for the same patient or response to treatment is currently not available.

Identification of the molecular signature of allograft rejection was the basis for the development of the molecular microscope, a diagnostic tool based on microarray analysis of EMB [58]. Researchers pooled individual molecular transcript data into pathway-driven subgroups called “pathogenesis-based transcripts” and analyzed them using machine learning bioinformatics approaches, resulting in the development and validation of classification algorithms called “classifiers” based on the expression levels of numerous genes [59]. Therefore, the molecular profile of the new biopsy is analyzed and compared with the profiles of all previously analyzed biopsies that are available in a database, which constitutes a “reference set.” As a result, the new biopsy is assigned a probability score belonging to a given diagnosis and enriches the reference set. The molecular test not only explores the probability of rejection, but also explores graft injury through dedicated transcripts and classifier [60]. The molecular microscope, initially developed in the academic setting at the University of Alberta, is now distributed by Thermo Fisher Laboratories. Its use in routine clinical practice in the United States has recently been validated by the FDA.

The advantages and disadvantages of the molecular microscope are common to many molecular diagnostic tests. Obtaining quantifiable data and objectivity are emphasized in the face of the “perfectible” reproducibility of pathologists [61]. The initial financial considerations of microarray analysis are no longer relevant, as costs continue to decrease. In the United States, the analysis is currently performed on two platforms, a factor that allows reproducibility and quality control and reasonable turnaround times for results (48 h after receipt of results according to the website). However, there are uncertainties concerning the representativeness of the sample submitted to the molecular microscope. Indeed, the procedure does not include a histological control of the fragment intended for molecular analysis. Therefore, cardiac transplant pathologists are confronted daily with these problems of representativeness of EMB: Non-representative biopsies in the case of pericardial transfixing biopsy, fibrin clusters, or biopsy in an anterior biopsy site [6]. The potential focal nature of rejection is well known, varying according to the biopsy fragments obtained during the same procedure. This problem is even more significant if we are interested in cardiac fibrosis or myocyte injury [62]. It is interesting to note that molecular analysis could, at least in theory, be a solution to these multifocal/representational issues during rejection. However, there are no studies that have compared the molecular profiles of multiple biopsies taken during the same diagnostic procedure. Similarly, “injury” transcripts and cardiac transcripts have scarcely been confronted with morphological analysis [60].

### 3.2. Restricted Molecular Signatures

While pan-transcriptomic studies have led to major advances in the field of transplant pathophysiology, many authors have questioned the relevance of restricted transcriptomic signatures, derived from genome-wide data, for the diagnosis of rejection in daily clinical practice. Major technological advances have allowed the exploration of 10 to several hundreds of transcripts from formalin-fixed paraffin-embedded (FFPE) tissue. The main advantages of using FFPE tissue are the ability to perform histopathology and the retrospective access to the blocks of inclusions, which allows longitudinal studies, crucial in heart transplantation (pre-injury diagnosis or response to treatment). Until recently, the main limitation of these techniques had been the relatively small number of targeted genes, which constrained discovery studies and the development of clinically relevant tools. As an example, our group has applied reverse transcription-multiplex ligation-dependent probe amplification (RT-MLPA), a multiplex RT-PCR technique to heart transplantation [63]. We could study the relative expression of 14 cardiac transcripts in 180+ EMB without rejection, with AMR, or with ACR. Our choice of transcripts of interest (endothelial, the interferon-gamma pathway, cytotoxicity, macrophagic, and NK cell transcripts) was driven by our previous microarray work, selecting the most differentially expressed transcripts in cardiac rejection [52]. This bioinformatics tool allowed us to develop two consecutive algorithms for molecular diagnosis, rejection versus non-rejection, then in the case of rejection, antibody-mediated versus cellular rejection. Examples of the profiles are shown in Figure 1. There was 92.2% agreement with the histopathologic diagnosis for the presence or absence of rejection and 79.2% agreement for the type of rejection [63].

The nCounter^®^ system allows the specific capture and counting of nucleic acids (mRNA, DNA, and miRNA) [64]. The nCounter^®^ allows targeting up to 800 genes organized in the form of a “panel” of genes. The particularity of the nCounter^®^ relies on the way it targets mRNA through a fluorescent “barcode” system. Each barcode is attached to a unique probe specific to a targeted mRNA, which is counted individually and directly without any amplification step. The probes need only 100 nucleotides to recognize their specific target. This sensitivity makes the nCounter^®^ particularly efficient on samples with degraded nucleic acids and/or in low-quantity samples, such as FFPE tissue. It was recently shown in heart transplantation that a signature of 34 transcripts had a better diagnostic performance than C4d and DSA, and correlated with endothelial activation assessed by electron microscopy [65].

The relevance of nCounter^®^ technology in solid organ transplantation has been underlined by numerous studies [66]. In the field of kidney transplantation, different teams have published restricted transcriptional signatures of rejection, but also of graft infections or BK virus nephritis and have shown their validity in a diagnostic approach [67]. This has led various groups to work on the development of a gene panel dedicated to transplantation, namely the Banff Human Organ Transplant (B-HOT) panel [67,68]. This multi-organ panel of 770 genes explores numerous molecular pathways of immuno-inflammation, infection, and tissue damage, among others and is being tested in the clinical field.

### 3.3. The Place of Molecular Biology

Confrontation with molecular data could improve the diagnosis of rejection and question the histopathological classifications of rejection. For example, as early as 2005 the Stanford team had shown that biopsies with mild or moderate ACR (grade 1A, 1B, and 2 according to the 1990 ISHLT classification), which had been included in the 1R category of the 2004 revised ISHT working formulation, significantly differed from a molecular point of view [54]. Pangenomic analysis revealed that biopsies graded 1A were closer to biopsies without rejection, while biopsies graded 1B and 2 were closer to proven moderate rejection 3A. Therefore, the current grade 1R includes rejections that are heterogeneous from a molecular point of view and may have distinct long-term prognoses. Regarding AMR, we showed that pAMR1 grade, suspected AMR, also comprised EMB with variable molecular activity [52]. An important point was that cases classified as pAMR1(I+), immunopathological AMR, had low molecular activity close to cases without rejection, whereas cases classified as pAMR1(H+), histopathological AMR, had molecular activity similar to those observed in pAMR2 and pAMR3. These findings highlight the importance of the histological signs of microvascular inflammation for the diagnosis of cardiac AMR. This led us to reexamine the 2013 pAMR classification, in terms of the diagnostic cut-off value of microvascular inflammation (MI), but also of MI quantification. This quantification of MI is absent from the 2013 pAMR classification, whereas it is largely accounted for in the Banff classification for renal allograft (the glomerulitis [g] and peritubular capillaritis [cpt] scores) [69]. When we studied the molecular profiles of cardiac AMR with semi-quantitatively graded MI, we found that the intensity of cardiac MI severity strongly correlated with molecular activity. Interestingly, high MI grade is almost consistently associated with high-intensity DSA and significantly associated with cardiac dysfunction [53]. Therefore, this MI score in heart transplantation seems to provide information that is complementary to the 2013 pAMR classification (Figure 2).

## 4. Integrated Bio-Histomolecular Diagnosis of Rejection

A key challenge remains to combine various sources of information in an integrated and more accurate diagnosis. Importantly, the evolution of bioinformatics techniques has contributed to the advancement in searching and predicting biomarkers, pathways, and new target drugs that allow a more precise and less invasive diagnosis [70].

In Figure 3, we proposed how non-invasive and tissular (i.e., myocardial samples) biomarkers of rejection may be associated in clinical practice to redefine the monitoring and diagnosis of rejection after heart transplantation. For asymptomatic patients, the individual stratification of the risk of rejection based on clinical variables, preferably integrated into an individual risk score and non-invasive biomarkers of rejection would classify the situation as low or high risk of rejection. For high-risk situations, an EMB should be performed. For symptomatic patients with clinical suspicion of rejection, an EMB should always be performed.

Once the EMB is performed, on top of standard pathological evaluation, a molecular evaluation may produce valuable information, particularly when there is an uncertain diagnosis, discrepancies between the clinical presentation and pathology (e.g., biopsy-negative rejection), or de novo DSA without pathologic features of AMR. The molecular diagnosis is seen as a companion tool and should not replace pathological evaluation. Finally, integrating pathology with molecular data and non-invasive biomarkers of graft injury may allow the analysis of the molecular activity of rejection and ongoing subclinical allograft injury. This information may be of high interest when discussing the treatment of subclinical rejection. The interest in an integrated bio-histomolecular diagnosis of rejection should be evaluated in dedicated prospective studies, both concerning the patient’s prognosis and the treatment of allograft rejection.

## 5. Conclusions

Major advances in the field of non-invasive and tissular biomarkers of cardiac allograft rejection have paved the way for a paradigm shift in the monitoring and diagnosis of cardiac allograft rejection. An integrated bio-histomolecular diagnosis of allograft rejection combining non-invasive biomarkers of allograft injury, tissue molecular activity, and pathology may have a major impact on the management of heart transplant recipients.

## Figures and Tables

**Figure 1 biomolecules-12-01135-f001:**
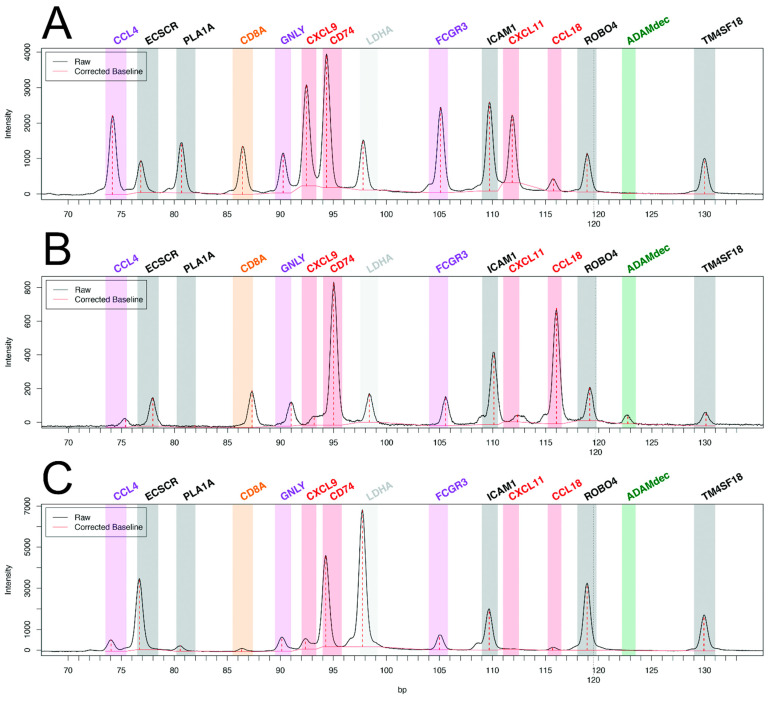
Molecular profiles of antibody-mediated rejection (**A**) with high expression of CCL4, PLA1A, GNLY, CXCL9, FCGR3, and CXCL11, and of acute cellular rejection (**B**) with high expression of ADAMdec and CCL18 as compared with non-rejecting control endomyocardial biopsy (**C**). Adapted from Adam et al. [63].

**Figure 2 biomolecules-12-01135-f002:**
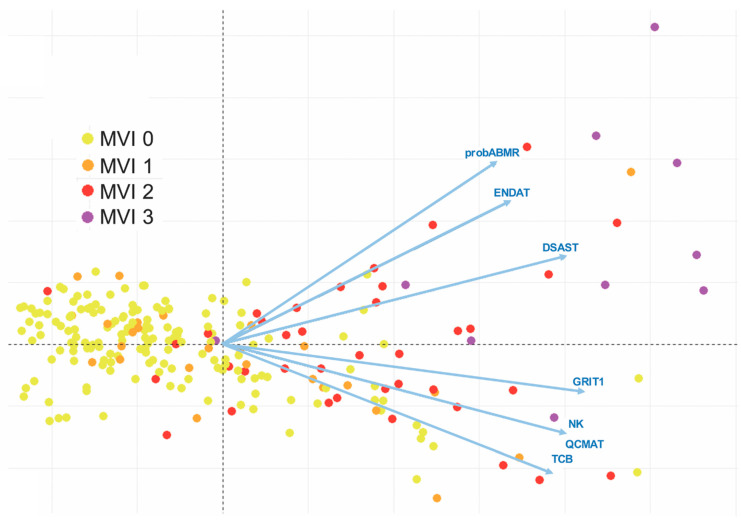
Correlation of molecular profiles with the microvascular inflammation (MVI) score—microarray analysis. Unsupervised principal component analysis (PCA) of MVI based on molecular profiles. Each dot represents one of the 182 endomyocardial biopsies (EMB) used for the microarray analysis, which are colored according to their MVI scores and distributed by their molecular features, as determined by the PCA. The PCA was based on six pathogenesis-based transcripts (DSAST: Donor-specific antibody transcripts; ENDAT: Endothelial transcripts; GRIT1: Interferon-gamma transcripts; NK: Infiltration of natural killer cells; QCMAT: Infiltration of macrophages; TCB: Infiltration of T cells), and one classifier score (probABMR). The extent of MVI was graded semi-quantitatively according to the percentage of myocardial area with MVI: MVI 0 represents 0% (negative MVI); MVI 1 is 1–10% (minimal MVI); MVI 2 is 11–50% (focal MVI); and MVI 3 represents > 50% (diffuse MVI). Areas of previous biopsy, replacement fibrosis, foci of ischemia, quilty lesions, and sites in close contact with foci of cellular rejection were excluded for the evaluation of MVI.

**Figure 3 biomolecules-12-01135-f003:**
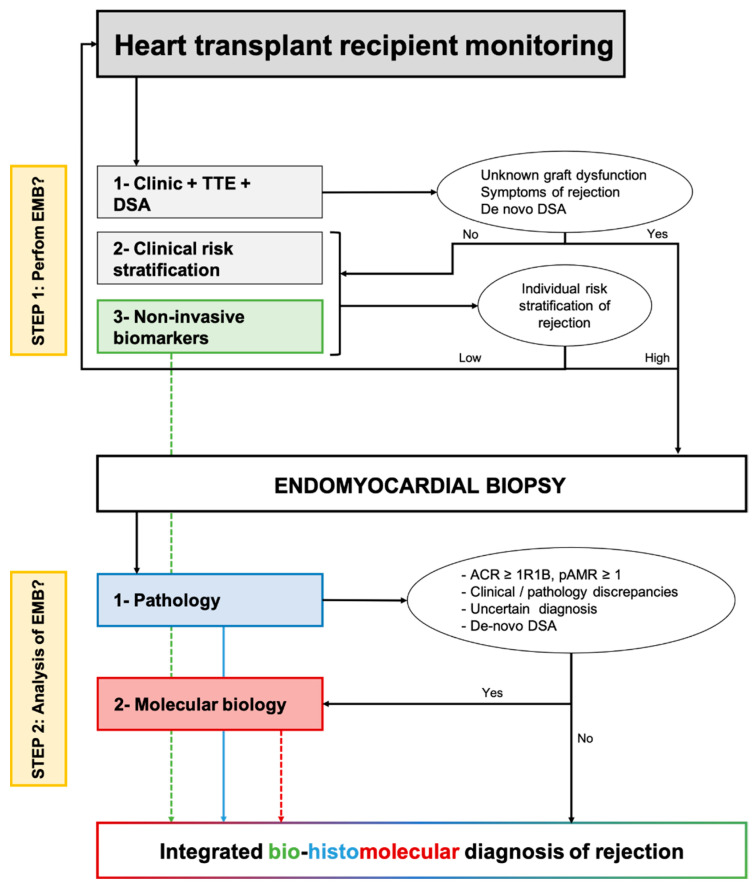
Proposed integration of non-invasive biomarkers and tissue-biomarkers to optimize the monitoring and diagnosis of cardiac allograft rejection. The first step relies on clinical and non-invasive biomarkers to stratify individually the risk of rejection and minimize the number of protocol endomyocardial biopsies. The first corresponds to the use of molecular biology as a companion tool of pathology to refine the diagnosis of allograft rejection. ACR: Acute cellular rejection; AMR: Antibody-mediated rejection; DSA: Donor-specific antibody; TTE: Trans-thoracic echocardiography.

**Table 1 biomolecules-12-01135-t001:** Proposed steps to develop and validate a non-invasive biomarker of allograft rejection.

#1—Biological plausibility		
#2—Technical aspects	Reproducibility	
	Stable over time	
	Known kinetics	
#3—Association with outcome	Case-control studies	
	Unselected patients	Prospective observational studies, cross-sectional design
		Prospective observational studies, longitudinal design
#4—Independent association with the outcome	Multivariate analyses including previously validated risk markers/factors of the outcome	
#5—Improved risk stratification	The addition of the new biomarkers on top of standard variables improves risk stratification (discrimination, calibration)	
#6—Interventional trial	Randomized clinical trial	
#7—Biomarkers vigilance	Long-term monitoring	

## Data Availability

The data generated and/or analyzed during the current study are not publicly available but are available from the corresponding author on reasonable request.

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
