# Peer review of "A Review of Biomarkers of Cardiac Allograft Rejection: Toward an Integrated Diagnosis of Rejection"

_biomolecules, 2022, doi:10.3390/biom12081135_

Round 1
Reviewer 1 Report
This article by Coutance et al is very well written and is indeed highly recommended for publication in the magazine. However, in order to slightly improve this article, I would make a minor revision. I would introduce in Table 1, an additional component of computational analysis or prediction of rejection biomarkers and/or design of new pathways and treatment drugs, as an example, I suggest naming this interesting recent article: Computational Prediction of Biomarkers, Pathways, and New Target Drugs in the Pathogenesis of Immune-Based Diseases Regarding Kidney Transplantation Rejection. Alfaro R, et al. Front Immunol. 2021 Dec 15;12:800968. doi: 10.3389/fimmu.2021.800968.
Author Response
Dear reviewer,
Thank you for your time and consideration.
Please find below a point-by-point answer to your remarks.
This article by Coutance et al is very well written and is indeed highly recommended for publication in the magazine.
We thank Reviewer #1 for these positive comments.
However, in order to slightly improve this article, I would make a minor revision. I would introduce in Table 1, an additional component of computational analysis or prediction of rejection biomarkers and/or design of new pathways and treatment drugs, as an example, I suggest naming this interesting recent article: Computational Prediction of Biomarkers, Pathways, and New Target Drugs in the Pathogenesis of Immune-Based Diseases Regarding Kidney Transplantation Rejection. Alfaro R, et al. Front Immunol. 2021 Dec 15;12:800968. doi: 10.3389/fimmu.2021.800968.
We thank Reviewer #1 for this proposition. We agree with Reviewer #1 that computational analysis and the the evolution of bioinformatics techniques are particularly important to integrate the data from various sources in a unique and integrated diagnosis.
However, since Table 1 focused only on the development on non-invasive biomarker, we prefered adding the suggested reference in a more general paragraph.
We added the following sentence at the beginning of the chapter "Integrated bio-histomolecular diagnosis of rejection":
"A key challenge remains to combine various sources of information in an integrated and more accurate diagnosis. Importantly, the evolution of bioinformatics techniques has contributed to the advancement in searching and predicting biomarkers, pathways, and new target drugs that allow a more precise and less invasive diagnosis [70]."
Reviewer 2 Report
this review about biomarkers and cardiac allograft rejection is well written and give to the readers a complete knowledge on cardiac heart transplant field. This review also elucidate a possible future for an endmyocardial biopsy diagnosis integration with some new biomarkers.
There are only two things to complete this review:
1- please introduce in the paragraph 2.2 non invasive biomarkers reflecting the inflammatory......5 or 6 lines refer to the last research about extracellular vesicles as non invasive biomarkers for allograft rejection citing for example Kennel et al. 2018 and Castellani et al. JHLT 2020.
2- in the section of miRNA and tissue molecular biology introduce also the paper of Di Francesco et al., JHLT 2018 where the authors used a panel of miRNA to identify the different types of rejections in heart transplant patients
with these two integration the review will be really complete and cover almost everything known about heart transplant
Author Response
Dear reviewer,
Thank you for your time and consideration and for your positive comments.
Please find below a point-by-point answer to your remarks.
1- Please introduce in the paragraph 2.2 non invasive biomarkers reflecting the inflammatory......5 or 6 lines refer to the last research about extracellular vesicles as non invasive biomarkers for allograft rejection citing for example Kennel et al. 2018 and Castellani et al. JHLT 2020.
We agree that extracellular vesicles are promising non-invasive biomarkers of rejection that should be included in the current manuscript. We therefore added a brief paragraph including the two important references.
“Recent research suggests that circulating extracellular vesicles may be a valuable non-invasive biomarker of rejection. They are raising considerable interest as they are easily detectable in blood and contain a specific set of nucleic acids, proteins, and lipids reflecting pathophysiologic conditions. Two recent retrospective case-control studies reported promising results that should be further evaluated and validated in unselected longitudinal cohorts [45,46].”
2- In the section of miRNA and tissue molecular biology introduce also the paper of Di Francesco et al., JHLT 2018 where the authors used a panel of miRNA to identify the different types of rejections in heart transplant patients
We agree with Reviewer #2 that the study from Di Francesco et al. is an important study in the field. However, we initially decided not to quote this reference since authors analyzed the miRNA signature on cardiac tissue and did not evaluate the interest of miRNA as a non-invasive biomarker of rejection.
We added the reference in the introduction section of the miRNA paragraph:
“Several studies have reported a potential causative role of miRNAs in the pathophysiology of cardiac allograft rejection and distinct miRNA profiles in EMB from patients with or without rejection [39–41].”
Reviewer 3 Report
I have read with great interest the review by Coutance et al. In their excellent work they summarize the current knowledge about the biomarkers (either invasive or non-invasive) used in the field of rejection of heart allografts. Review has high inner coherence, is very well written and besides several typos (listed bellow) I also list some minor comments that may improve already a brilliant text.
Minor commnets:
1. Line 16 (abstract) and line 37 (part 1): Please, insert „increased“ to make the sentences sound better, „…with INCREASED morbidity and mortality.“
2. Line 38: Please specify whether by sentence „ACR has declined significantly over time“ you mean that ACR decreases over the time from transplantation/operation itself or whether you mean that ACR prevalence decreases over the time from the initiation of the transplantation programs due to the improvements in immunosuppression
3. Line 55: Sentence „…particularly concerning the acceptability of the strategy by patients.“ needs to be more clarified as our patients from our transplant center does not have problem to accept the protocol EMB performance. I agree that EMB does not fulfil screening test criteria by WHO, but selecting this particular criterion does not seem to me to be the biggest issue
4. Line 223-231: The end of the microRNAs section seems to be „suddenly ended“. Authors are mentioning that several studies were reported to perform miRNA analysis in myocardial tissue, however, they comment the results of only one study and there is a lacking reference to it. There are also other studies (e.g. Cells | Free Full-Text | Identification of a Diagnostic Set of Endomyocardial Biopsy microRNAs for Acute Cellular Rejection Diagnostics in Patients after Heart Transplantation Using Next-Generation Sequencing (mdpi.com)) focusing on miRNA analysis in EMB that authors do not comment at all. I suggest that authors either update the list of miRNA-EMB studies, or they can refer to the recent review focusing solely on miRNAs as markers of heart transplant rejection (MicroRNAs as theranostic markers in cardiac allograft transplantation: from murine models to clinical practice - PubMed (nih.gov)) to direct the reader for further reading, if they want to avoid unnecessary expansion of the text.
5. Line 256-265: I suggest moving this whole paragraph at the beginning of the next section, i.e. immediatelly after the heading „3. Molecular biology as a companion tool to refine the diagnosis of rejection“.
6. Line 426. Term „tissular“ shall be explained.
Typos:
· Line 189: “to monito“ shall be „to monitor“
· Line 190: BEM shall be EMB
· Line 425 and 441 and 457: Please unify the use of hyphens: either „bio-histomolecular“ or „bio-histo-molecular“ both in the text and title
Author Response
Dear reviewer,
Thank you for your time and consideration.
Please find below a point-by-point answer to your remarks.
I have read with great interest the review by Coutance et al. In their excellent work they summarize the current knowledge about the biomarkers (either invasive or non-invasive) used in the field of rejection of heart allografts. Review has high inner coherence, is very well written and besides several typos (listed bellow) I also list some minor comments that may improve already a brilliant text.
We thank Reviewer #3 for these positivie comments.
Minor commnets:
1. Line 16 (abstract) and line 37 (part 1): Please, insert „increased“ to make the sentences sound better, „…with INCREASED morbidity and mortality.“
We thank Reviewer#3 for this precision and modified the manuscript accordingly.
2. Line 38: Please specify whether by sentence „ACR has declined significantly over time“ you mean that ACR decreases over the time from transplantation/operation itself or whether you mean that ACR prevalence decreases over the time from the initiation of the transplantation programs due to the improvements in immunosuppression
We acknowledge the lack of clarity of the sentence. We changed the sentence as proposed:
“While the prevalence of acute cellular rejection (ACR) decreased over the time from the initiation of the transplantation programs due to the improvements in immunosuppression [7,8] (…)”
3. Line 55: Sentence „…particularly concerning the acceptability of the strategy by patients.“ needs to be more clarified as our patients from our transplant center does not have problem to accept the protocol EMB performance. I agree that EMB does not fulfil screening test criteria by WHO, but selecting this particular criterion does not seem to me to be the biggest issue
We agree with Reviewer #3 and removed a part of the sentence:
"EMB as a screening test does not fulfill all the principles of screening recommended by the World Health Organization particularly concerning the acceptability of the strategy by patients."
4. Line 223-231: The end of the microRNAs section seems to be „suddenly ended“. Authors are mentioning that several studies were reported to perform miRNA analysis in myocardial tissue, however, they comment the results of only one study and there is a lacking reference to it. There are also other studies (e.g. Cells | Free Full-Text | Identification of a Diagnostic Set of Endomyocardial Biopsy microRNAs for Acute Cellular Rejection Diagnostics in Patients after Heart Transplantation Using Next-Generation Sequencing (mdpi.com)) focusing on miRNA analysis in EMB that authors do not comment at all. I suggest that authors either update the list of miRNA-EMB studies, or they can refer to the recent review focusing solely on miRNAs as markers of heart transplant rejection (MicroRNAs as theranostic markers in cardiac allograft transplantation: from murine models to clinical practice - PubMed (nih.gov)) to direct the reader for further reading, if they want to avoid unnecessary expansion of the text.
We agree with Reviewer #3 that these two important references were missing. Our aim was mainly to focus on miRNA as non-invasive biomarkers of rejection. We missed references concerning the potential role of miRNA profiling on myocardial samples (“invasive biomarker”). We added the two references suggested and introduce the systematic review to direct the reader for further reading.
"Several studies have reported a potential causative role of miRNAs in the pathophysiology of cardiac allograft rejection and distinct miRNA profiles in EMB from patients with or without rejection [39–42]. These studies have been summarized in a comprehensive review [43]."
5. Line 256-265: I suggest moving this whole paragraph at the beginning of the next section, i.e. immediatelly after the heading „3. Molecular biology as a companion tool to refine the diagnosis of rejection“.
We agree with Reviewer #3 and moved the whole paragraph at the beginning of the next section.
6. Line 426. Term „tissular“ shall be explained.
We defined tissular as suggested:
In Figure 3, we proposed how non-invasive and tissular (i.e., myocardial samples) biomarkers of rejection may be associated in clinical practice to redefine the monitoring and diagnosis of rejection after heart transplantation.
Typos:
- Line 189: “to monito“ shall be „to monitor“
- Line 190: BEM shall be EMB
- Line 425 and 441 and 457: Please unify the use of hyphens: either „bio-histomolecular“ or „bio-histo-molecular“ both in the text and title
Wethank Reviewer #3 for the identification of typos. We corrected them accordingly.